# Diagnostic Aspects of an Included Third Molar in an 88-Year-Old Patient: A Case Report and Literature Review

**DOI:** 10.3390/diagnostics12092082

**Published:** 2022-08-28

**Authors:** Mariana I. Miron, Ciprian T. Florea, Diana Lungeanu, Carmen D. Todea

**Affiliations:** 1Department of Oral Rehabilitation and Dental Emergencies, Faculty of Dentistry, “Victor Babes” University of Medicine and Pharmacy, 300041 Timisoara, Romania; 2Interdisciplinary Research Center for Dental Medical Research, Lasers and Innovative Technologies, “Victor Babes” University of Medicine and Pharmacy, 300041 Timisoara, Romania; 3Center for Modeling Biological Systems and Data Analysis, “Victor Babes” University of Medicine and Pharmacy, 300041 Timisoara, Romania; 4Department of Functional Sciences, Faculty of Medicine, “Victor Babes” University of Medicine and Pharmacy, 300041 Timisoara, Romania

**Keywords:** third molar, dental inclusion, elderly patient, pericoronitis, operculectomy, mandibular wisdom tooth

## Abstract

Included third molars in elderly patients are quite rare in dental practice, and therefore easily misdiagnosed, because these teeth are usually extracted in youth. Additional challenges to correctly diagnosing such a dental condition, and its associated complications, arise from frequent co-morbidities in elderly patients, and from difficult communication with the patient. We report a case of an 88-year-old female patient, who presented in the dental emergency room complaining of a discomfort caused by the sharp edges of her lower incisors, and requesting their extraction; the final diagnosis, of suppurated pericoronitis at tooth 48, was concluded based on the clinical elements and X-ray examination.

## 1. Introduction

Over the years, issues related to the third molar (commonly called the wisdom tooth) have been intensely investigated by researchers, from multiple perspectives, the most common being eruption issues, dental inclusion, extraction techniques, and their associated complications. Third molars usually erupt in the late teens up to the early twenties; in most cases, the eruption is accompanied by local and/or loco-regional complications, partially due to insufficient space in the dental arch [1]. According to Kautto, et al. [2], the peak age of third molar extraction ranges from 23 to 25 years. However, there are many situations in which the eruption of the third molar occurs in mature adult, or even elderly, patients, who are not aware that they still have an included third molar; this lack of awareness is mainly due to the absence of symptoms, rare and inconsistent dental treatments, or improper dental examinations. In such cases, patients present to the dental emergency service, due to pain and complications, with no obvious etiology.

It is acknowledged that dental treatment of the elderly can be difficult: on the one hand, if the senior patients do not receive the sought dentist’s understanding, they migrate from one dentist to another, until they find one they feel comfortable with; on the other hand, the elderly population has diminished interest in dental treatments, making dentist’s appointments only when the pain occurs and, quite often, after several unsuccessful artisanal treatments. Moreover, the healing potential is decreased in older patients, with increased risk of complications [3,4,5,6,7]. Phillips, et al. [8], reported age as a significant predictor of delayed clinical recovery after third molar surgery.

The aim of this report was to present a case of an included third molar becoming apparent in an elderly patient.

## 2. Case Report

An 88-year-old female patient presented in the dental emergency room of “Victor Babes” University of Medicine and Pharmacy from Timisoara, complaining of difficulties and pain in mastication and phonation; according to the patient, the problems were caused by her mandible frontal crowns. The patient’s mandible frontal teeth had extensive and irregular coronary destruction, therefore representing micro-irritation factors for the surrounding tissues (Figure 1). From the beginning of the consultation, the patient’s urgent request was for the dental roots of her frontal mandibular teeth to be extracted.

The patient was cooperative and in good mental condition and general health; she independently filled out and signed the medical questionnaire and the informed consent.

According to the medical letter she presented, her medical history revealed several underlying health conditions: fourth degree high blood pressure with very high risk; persistent atrial fibrillation, complicated by class 2 heart failure, as classified by the New York Heart Association’s (NYHA); moderate degenerative mitral regurgitation and calcified fibrosis aortic valve; and dyslipidemia. The patient had undergone a cataract surgery a year before. For all conditions, the patient was taking prescribed drug treatments.

The patient’s blood pressure was recorded within normal limits, as 125/70 mmHg, and her body temperature was 36.5 degrees Celsius.

During the exo-oral examination, we noticed a slight facial asymmetry with a reduced muscle tonus, and slightly fallen commissure towards the right side.

The anamnesis disclosed that the patient’s dental discomfort and medical condition had worsened two weeks before, showing: pain in the right hemi-mandible, radiating to the ear; hemi-mandible paresthesia in quadrant four; limited mouth-opening; and masticatory discomfort in the mandibular frontal area. Loco-regional examination revealed a slight right sub-mandible lymphadenopathy, but the patient’s general condition was good. The patient had self-administered non-steroidal anti-inflammatory drugs (NSAID) for three days, and the pain had subsided, but the hemi-mandible paresthesia did not disappear. The patient also told us that she was wearing mobile prostheses, but that she had stopped using them about two weeks previously, when the pain and paresthesia started.

An intra-oral examination was conducted: at the mandible level, the patient presented only the dental roots of the incisors with different degrees of destruction and sharp edges, thus causing mastication difficulties (Figure 1); at the level of 31, 32, and 41, the carious process had completely destroyed the teeth’s crowns; at the level of 41, the crown showed loss of dental hard substance of atypically retentive form. All four incisors actually acted like irritating spines for the patient, and contributed to the micro-irritation of the surrounding tissue. Figure 1 also reveals the color and shape of the mandible’s edentulous ridge.

An erythematous and edematous zone was observed on the distal area of the right mandible ridge (Figure 2a). That spot was painful on manual palpation, and a small purulent secretion with a relatively high consistency was expressed (Figure 2b). Nevertheless, no dental root or hard structure was detected, either on manual palpation or on the periodontal probe.

Based on the above clinical symptoms, we resumed our discussion with the patient, and employed targeted questions to identify the etiological factors of her discomfort. We received negative answers regarding any recollection of a molar previously extracted from that area, any included wisdom tooth, or any other symptomatic episode after the prosthesis had been made.

In order to improve the clinical symptoms, to control the local infection, and to establish a certain diagnosis, the following steps were performed:selective preparation of the cutting edges of the lower incisors (coronary and radicular), so that they no longer micro-irritated the surrounding soft tissue;anti-inflammatory (Ibuprofen, 400 milligrams) and antibiotic (Amoxicillin, 1 g, two times per day, for seven days) medication was prescribed;the patient was referred for a panoramic radiography;the patient was rescheduled for the next morning.

At the next-day appointment, the X-ray panoramic image revealed the presence of the third molar included in the right hemi-mandible bone (tooth 48), vertically positioned as shown in Figure 3.

Based on all clinical elements and the X-ray examination, we excluded diagnoses of superinfected root residue, superinfected residual cyst, or odontogenic tumor, and the final diagnosis of suppurated pericoronitis at tooth 48 was concluded.

The patient was under antibiotic therapy, so an operculectomy was subsequently performed, as the elective emergency treatment for suppurated pericoronitis (Figure 4).

Summing up, the following operations were performed during the second visit to the dental emergency room:loco-regional anesthesia at the Spix’s spine, using Ubistesin carpule, 1 mL solution 1:200,000, 3M ESPE, Deutschland GmbH., Car-Schurz-Straвe 1, 41,453 Neuss Germany;operculectomy using a surgical carbon steel No. 15 blade, SMI AG, Steinerberg 8, 4780 St Vith/Belgium (an incision was made at the level of the entire area mucosa, and the operculum was removed with surgical forceps);appropriate drainage was performed, using sterile solutions (normal saline and hydrogen peroxide) to irrigate the pericoronal space;continuation of antibiotic therapy was recommended, accompanied by anti-inflammatory medication for pain control, and use of chlorhexidine solution for mouth-washing;the patient was referred to a dento-alveolar surgery specialist for the extraction of the molar 48 after the acute phase was over.

## 3. Literature Review and Discussion

Making a correct diagnosis is one of the most important stages of any dental treatment. Correctly diagnosing the included third molar in an elderly patient can be a challenge, because these teeth do not usually survive into old age, and such a diagnosis is very rare: the third molar occupies a special position inside the dental arch, is the last to erupt, and is the first to be extracted [9]. On the other hand, the actual practice is to not remove these teeth in adolescence, if there is no underlying medical reason to do so, and therefore such cases are expected to become increasingly prevalent in mature patients.

The proportion of elderly in the active population is increasing, as are the associated orodental complications (e.g., a retained third molar previously overlooked) [1,10]. Elderly patients can experience reduced symptomatology, but face additional medical complications generated by poor communication and old-age comorbidities [7,11].

The literature specifically reporting on this particular subject is scarce. Table 1 summarizes the papers that we found, on the clinical symptomatology and complications, treatment options, and management issues related to the third molar in the elderly. A literature search was carried out, using the following electronic databases: MEDLINE via PubMed; Web of Science; EMBASE; SCOPUS; Science Direct; and Google Scholar. The keyword combination was: “(third OR wisdom) AND molar AND elderly AND (included OR impacted OR unerupted) AND (diagnosis OR symptomatology OR symptoms OR status)”. The review was conducted in July 2022, and was limited to articles published from 2000 to 2022. The eligibility criteria followed the qualitative review tool PECOS/PICOS (Population, Exposure/Intervention, Comparison, Outcomes, Studies) [12], applied as follows: (P) adult and elderly subjects; (E) presence and status of third molar on the dental arch; (C) absence of third-molar characteristic symptomatology; (O) association between third-molar symptomatology and its status on the dental arch; (S) observational studies/cross-sectional studies/case reports. The exclusion criteria were: (a) studies on children, adolescents, or young people (younger than 45 years); (b) studies that did not correlate third-molar status and symptomatology; (c) studies with full-text not available; (d) studies not in English and/or with no English translation.

Di Lauro, et al. [13], communicated a case report of a 64-year-old male with lower third molar inclusion in which paresthesia was the dominant pre-operative symptom of pericoronitis. The patient had permanent paresthesia of the left hemi-lip, which was accentuated during inflammatory episodes, and odontotomy (a surgical treatment) was performed after the proper evaluation. In our case, the 88-year-old female presented in the dental emergency room complaining of discomfort caused by the sharp edges of the lower incisors; she had had paresthesia of the right mandibular side for two weeks, a condition which did not subside under anti-inflammatory medication. Due to possible future complications, we recommended surgical treatment.

Yamaoka, et al. [14], studied the influence of ageing, and the contact of the adjacent teeth, on purulent inflammation associated with completely impacted lower third molars. They observed that non-contact adjacent teeth associated with purulent inflammation in older patients mostly occurred after 45 years of age. Their study revealed that the incidence of symptoms did not differ in patients of various age groups, other than 41–50 years of age, but that patients with inflammation were older than those with pain only. Their results indicated age as a strong predictor of the purulent inflammation associated with completely impacted lower third molars, regardless of their contact with adjacent second molars. A similar situation was present in our case, where molar 48 was completely included, with no adjacent second molar, but was associated with well-localized purulent inflammation and minor pain sensation. The hemi-paresthesia could also have reduced the local pain.

**Table 1 diagnostics-12-02082-t001:** Studies on the clinical symptomatology and complications associated with the third molar in the elderly (order of relevance to the present case report).

Authors	Year	Paper Ref.	Study Design	*n*, N, Age, Gender	3M Symptomatology	3M Status
Di Lauro, et al.	2021	[13]	Case report.	*n* = 1, 64 y, M	Severe paresthesia; generalized periodontitis stage IV grade C.	Included in horizontal position.
Yamaoka, et al.	1997	[14]	Cross-sectional.	*n* = 26, N = 800 34 ± 10.7 y, M, F	Pain and purulent inflammation.	Completely impacted.
Elter, et al.	2005	[15]	Cross-sectional.	*n* = 2209, N = 6967 52–74 y M, F	Inflammation; redness; suppuration and gingival tissue; changes in the third molar region, including gingival hyperplasia.	Visible.
Moss, et al.	2009	[16]	Cross-sectional.	*n* = 2035, N = 6793 62.4 ± 5.6 y M, F	Periodontal pathology; inflammation.	Visible.
Chou, et al.	2017	[17]	Cross-sectional split-mouth study.	*n* = N = 70 >45 y, M, F	Caries and periodontal disease on the adjacent molar.	Retained; unilaterally erupted 3Ms.
Kim, et al.	2017	[18]	Retrospective analysis of clinical records.	*n* = 2883 impacted 3Ms, N = 1109 29.0 ± 10.2 y M, F	Pericoronitis, dental caries, and root resorption on adjacent molar; infection.	Retained; impacted.
Ventä, et al.	2015	[6]	Cross-sectional.	*n* = 56, N = 293 79 ± 3.8 y M, F	Caries and periodontal pathology; mesio-angular inclination; cyst; resorption; ankylosis; over-eruption.	Impacted or deeply impacted; included.
Ventä, et al.	2018	[19]	Cross-sectional.	*n* = 2653, N = 6005 30–93 y M, F	Disease-free, asymptomatic caries; periodontal pathology on the adjacent molar.	Retained; impacted.
Taheri, et al.	2020	[20]	Case report.	*n* = 1, 94 y, F	Suppurative pericoronitis.	Included.
Anyanechi, et al.	2019	[21]	3-year longitudinal study.	*n* = 287, N = 1261 >50 y, M, F	Caries, gingivitis, pericoronitis; symptomatic burden of impacted 3Ms on the adjacent second molars.	
Baensch, et al.	2017	[7]	Case-control matched-pair analysis.	N = 127 pairs A: >65 y B: 15–20 y M, F	Pericoronitis; caries; resorption.	Retained; impacted.
Kim et al.	2021	[5]	Editor’s opinion.	-	-	-

Abbreviations: 3M, third molar; F, female; M, male; m ± SD, mean ± standard deviation; *n*, number of cases; N, total number of subjects; y, years.

Frequent co-morbidities in elderly patients influence the reported symptoms, and therapeutic and surgical evolution. Elter, et al. [15], reported a significant association between the visible third molar and the periodontal disease (as measured by probing depth and gingival bleeding) on the adjacent second molar in subjects with an average age of 62 years, who were enrolled in the Dental arm of the Atherosclerosis Risk in Communities (ARIC) study, namely DARIC. In the present case, the molar 48 did not actually erupt; however, the crown became more superficial, due to long absorption of the mandibular alveolar bone. The local pressure of the removable prostheses began to irritate the underlying mucosa against the crown. The apparent periodontal disease around the tooth may have further amplified the process. Moss, et al. [16], also reported that, for middle-aged and older participants, a visible third molar was significantly associated with a pattern of more extensive periodontal disease, compared to those without a visible third molar; their subjects were also from the DARIC study.

Chou, et al. [17], provided evidence for the association of third molars with the risk of caries and periodontal disease in the adjacent second molars of elderly patients. In contrast to the subjects in the DARIC study [15,16], these patients were relatively healthy, and had no smoking history. Therefore, their results confirmed that retained erupted third molars cause lesions to adjacent second molars in the elderly, even when controlling for co-variates, such as systemic diseases and smoking habits.

Furthermore, in a retrospective analysis of the clinical records, Kim, et al. [18], found a significantly higher risk of serious pathologic conditions (i.e., infection requiring antibiotics or surgical incision and drainage of the abscess, and hospital admission) among older patients with third molars.

Venta, et al. [6], studied 293 panoramic radiographs, out of which 112 (38%) were from totally edentulous subjects. Their results revealed that signs of periodontal pathology of third molars were more frequent in the mandible compared to the maxilla, with pathology present in every third molar in that sample. The authors recommended the extraction of these teeth at a younger age. Venta, et al. [19], subsequently performed a study on disease-free third molars in the adult Finnish population, and their results showed that third molars at the cervical or apical levels (or situated deeper in the bone) were more likely to remain disease-free than those at the occlusal level. The case we report corroborated these results: the included molar was on an edentulous mandible ridge, and remained disease-free.

According to Taheri, et al. [20], extractions of third molars in elderly patients are not common, and should be thoroughly evaluated, due to possible intraoperative complications (mandibular fracture or trauma of adjacent teeth), and postoperative complications (bleeding and infection). Nevertheless, when infection or other oral pathology is associated with third molars, a surgical approach remains the best therapeutic decision. Anyanechi, et al. [21], also recommended the extraction of symptomatic teeth, when necessary, in patients over 50 years.

Baensch, et al. [7], advised careful management of the third molar extraction in elderly patients, due to the higher risk of surgical complications and the longer time needed for intervention. It is acknowledged that healing potential is decreased in older patients, and also that the risk of complications increases with age [5].

The clinical observations in our case were consistent with those of the above authors. The 88-year-old female patient had multiple ailments and high-risk co-morbidities, but was under medical treatment, and in a well-balanced general condition. The first priority in the emergency room was to ensure a stable, well-controlled oral condition for the patient. Another important initial obstacle to surmount was to establish effective communication, in order not only to carry out the complete anamnesis, but also to gain the trust and collaboration of the patient. As she started to feel pain, the discussion was challenging, and the gathering of information was difficult. An elderly patient’s perception of pain may be different to a young patient’s, depending on their general and mental health status [22,23]. The information required in the dental consultation may be obtained through well-targeted questions, in parallel with careful and timely consultation.

The sensation of paresthesia of the hemi-mandible has been highlighted by researchers as a particular diagnostic element for lower third molar inclusion [13]. However, in our case, it was not reported by the patient, even though she had experienced it for two weeks, but was detected by the dentist during the detailed anamnesis. We emphasize the importance of focusing the emergency therapy on reducing pain and controlling infection in such an eruptive disorder, as the evidence from randomized controlled clinical trials recommends [24].

When infection or other oral pathology is associated with the third molar in an elderly patient, the therapeutic decision of choice is the surgical approach, as recommended in the literature [19]. However, in our patient, the final extraction treatment could be performed only after the ending of the acute phase, and under close supervision of the general medical condition of the patient.

International guidelines for the management of the third molar [25,26] also state that third molar management must begin with a thorough medical and dental history, focusing on symptoms. Afterwards, a panoramic radiography is recommended, to make the final diagnosis when surgical intervention is being considered. These guidelines were applied in the management of the present case.

## 4. Conclusions

An unerupted and/or impacted third molar is diagnosed clinically by detailed history-taking and thorough clinical examination. A rigorous exo-oral and endo-oral examination is mandatory, even in dental emergency situations. Special attention must be paid to elderly patients, who may be unable to explicitly communicate their medical history and their symptoms of an included third molar, which may lead to a misdiagnosis. All the required elements of a patient’s clinical examination (inspection, palpation, percussion, auscultation) are needed for an accurate diagnosis, and for a successful treatment in the short and long term.

Summing up the present case, the 88-year-old female patient presented in the dental emergency room with atypical symptoms and requests. A detailed anamnesis and comprehensive endo-oral examination highlighted the presence of inflammation and of a suppurative process located distally on the right mandibular edentulous ridge. The panoramic radiography was the decisive paraclinical investigation for the correct diagnosis.

## Figures and Tables

**Figure 1 diagnostics-12-02082-f001:**
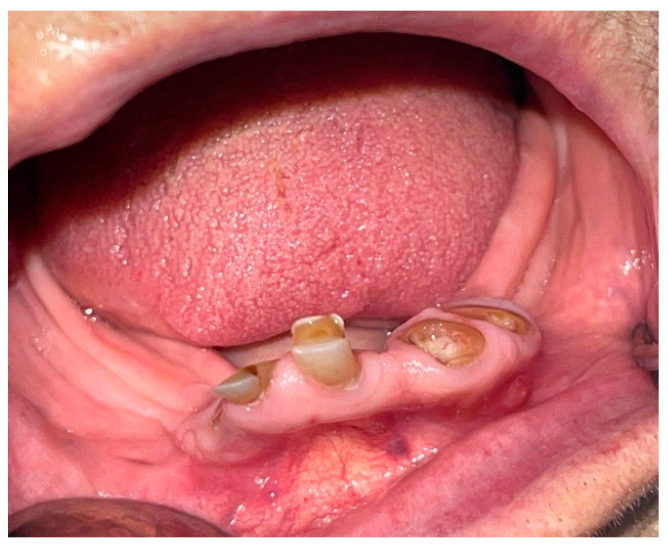
The features of the mandible incisors and the mandibular edentulous ridge.

**Figure 2 diagnostics-12-02082-f002:**
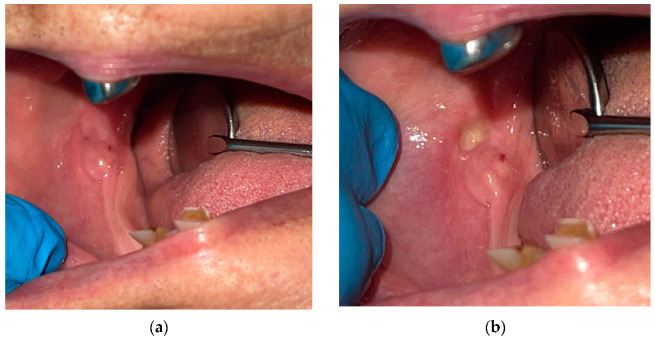
The distal area of the right mandible ridge: (**a**) erythematous and edematous area; (**b**) purulent secretion, expressed by manual palpation.

**Figure 3 diagnostics-12-02082-f003:**
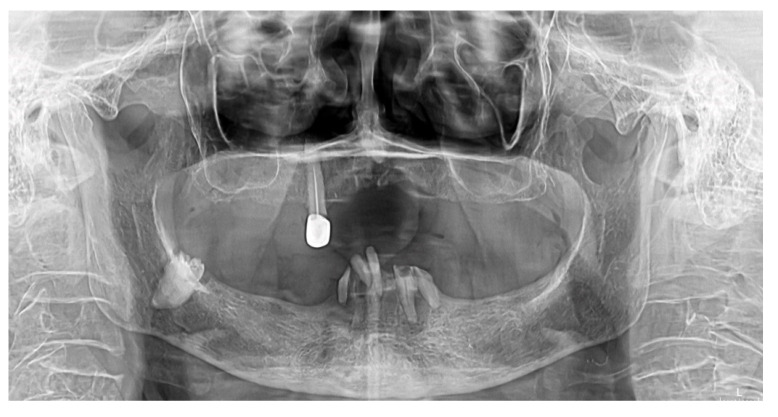
X-ray panoramic image of the 88-year-old female patient, showing the presence of the molar 48 included in a vertical position.

**Figure 4 diagnostics-12-02082-f004:**
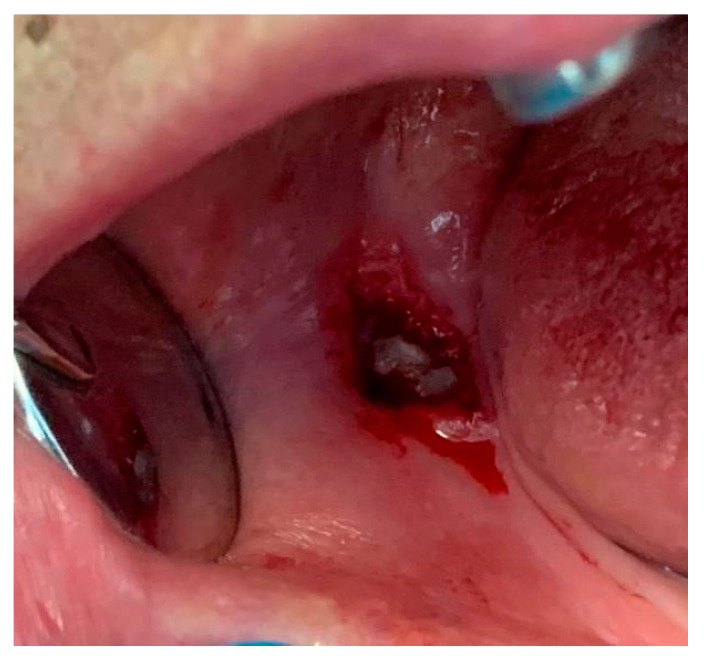
The affected area after the operculectomy.

## Data Availability

Not applicable.

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
