# Peer review of "Diagnostic Aspects of an Included Third Molar in an 88-Year-Old Patient: A Case Report and Literature Review"

_diagnostics, 2022, doi:10.3390/diagnostics12092082_

Round 1

Reviewer 1 Report

01

“We aim to increase the dentists' awareness about the elderly patients with the third molar included, who may present in the dental emergency room with atypical clinical symptoms and urgent treatment requirements, thus masking the real diagnosis and misleading the clinician in unnecessary or inappropriate treatment.”

The aim of this manuscript was to present a case of a third molar “eruption” in an elderly patient. Point. The rest of it is left for discussion.

02

The title is wrong. The tooth did not erupt. A part of the tooth crown became more superficial due to long absorption of the mandibular alveolar bone, until it reached the point that the local pressure of the removable prosthesis began to irritate the mucosa underneath against the crown. The apparent periodontal disease around the tooth could have helped in the process. The tooth did not move coronally, not at all.

03

There are some sentences in the text without reference to a previous study (or studies) in order to give evidence to their statements. Without references, these statements would be mere assumptions or allegations by the authors of the manuscript. Therefore, each of the following sentences need at least one reference to back up their statement:

“The proportion of elderly in the active population is increasing and so are the associated orodental complications (e.g., a retained third molar previously overlooked).”

“The elderly patient can experience reduced symptomatology, but brings additional medical complications generated by the poor communication and old-age commorbidities.”

04

“Table 1 synthesizes the papers on the clinical symptomatology and complications, treatment options, and management issues of the third molar in elderly we found.”

How was the process of search for these papers? Nothing is described.

05

A great part of the Discussion section consists of paragraphs describing other similar cases reported in the literature, without an actual discussion of the findings of the case presented by the authors.

06

Your conclusion is a synthesis of recommendations based on a literature review. Please conclude based on the findings of your case.

Author Response

Thank You for the thorough revision, comments, and suggestions for improving the material. Please find attached herewith the responses to the raised issues and suggestions.

Reviewer 2 Report

The importance of this study is in link-up with the identification of atypical clinical symptoms and the emergency treatment requirements of included and impacted third molar in senior patients.

The manuscript is in reference with the proposed subject in title.

The reference list of the manuscript contain 21 titles, is without inappropriate self-citations, and include 7 titles older than 5 years.

The case presentation and the analyzed date are written in proper way. The discussions are significant, and are suitable interpreted. All aspects regarding the figures/images and the table are appropriate, and they are easy to understand and to interpret.

The paper is clear, and has a good rate of significance. The presented data are important and sustain the conclusions.

The manuscript present scientific importance because the atypical clinical symptoms of impacted third molar in senior patients can mask the real diagnosis and can induce the effectuation of the inappropriate treatments.

The overall merit of the manuscript is good, and provide an advance to the current state of knowledge.

Author Response

Thank You for the appreciation of our work. We further improved the manuscript during the revision process.

Round 2

Reviewer 1 Report

The manuscript now seems to be suitable for publication.